# Diagnostic classification based on DNA methylation profiles using sequential machine learning approaches

**Marcin W. Wojewodzic**[1,2,3]*, **Jan P. Lavender**[3]

**1** Cancer Registry of Norway, Norwegian Institute of Public Health, Oslo, Norway, **2** Chemical Toxicology, Norwegian Institute of Public Health, Oslo, Norway, **3** University of Birmingham, Birmingham, United Kingdom

* Marcin.Wojewodzic@kreftregisteret.no

## Abstract

Aberrant methylation patterns in human DNA have great potential for the discovery of novel diagnostic and disease progression biomarkers. In this paper we used machine learning algorithms to identify promising methylation sites for diagnosing cancerous tissue and to classify patients based on methylation values at these sites. We used genome-wide DNA methylation patterns from both cancerous and normal tissue samples, obtained from the Genomic Data Commons consortium and trialled our methods on three types of urological cancer. A decision tree was used to identify the methylation sites most useful for diagnosis. The identified locations were then used to train a neural network to classify samples as either cancerous or non-cancerous. Using this two-step approach we found strong indicative biomarker panels for each of the three cancer types. These methods could likely be translated to other cancers and improved by using non-invasive liquid methods such as blood instead of biopsy tissue.

## Introduction

Epigenetics is the study of biochemical modifications, heritable through cell division, carrying information independent of DNA sequence [1]. This information has direct consequences for cellular phenotype and cell differentiation, including cancer formation. Although the biochemical modifications can vary for DNA, methylation signatures seem to be most optimal in terms of tradeoffs between variation and stability to provide data which can be used for disease diagnosis. This, together with recent breakthroughs and the decreasing cost of sequencing technology, is a probable reason that methylation is the branch of epigenetics that has progressed furthest in the last decade.

More importantly, clear evidence has emerged in recent years that aberrant DNA methylation plays a key role in many diseases, including cancer [2–4]. As a key epigenetic modification, this biochemical process can modulate gene expression to influence the cell differentiation which can possibly lead to cancer [2]. For cancer, aberrant methylation patterns seem to be a promising source of early diagnostic biomarkers. Mayeux and colleagues divide

**Data Availability Statement:** * The source code is available in GitHub repository: https://github.com/bazyliszek/methAI and in Zenodo: https://doi.org/

10.5281/zenodo.11530987 * All example datasets were used in previously published studies and are available. We used publicly available and anonymous data from The Cancer Genome Atlas and Harmonized Cancer Datasets in the Genomic Data Commons Data Portal (https://www.cancer.gov/tcga, https://portal.gdc.cancer.gov/) investigating kidney, prostate and bladder tissues (dataset https://portal.gdc.cancer.gov/legacyarchive/search/f) assayed in multiple, independent experiments using the Illumina 450k microarray. The clinical metadata and the manifest file of the samples were downloaded through the GDC legacy archive on 2019-10-19 and 2019-11-09. We used the Genomic Data Commons (GDC) data download tool (https://gdc.cancer.gov/access-data/gdc-data-transfer-tool) using the manifest files deposited on our GitHub.

**Funding:** The author(s) received no specific funding for this work.

**Competing interests:** The authors have declared that no competing interests exist.

the major types of biomarkers into: biomarkers of exposure (used in risk prediction) and biomarkers of disease used in the screening, diagnosis and monitoring of disease progression) [5]. Early diagnosis biomarkers are of particular interest as it has been shown that, for many cancer types, early detection is strongly correlated with the patient's chance of survival [6].

Finding methylation patterns in human DNA that are specific to different environmental exposures (e.g. cancerogenic substances) and clinical diagnoses would be a powerful tool to develop cost-efficient predictive assays which could guide clinicians and save lives. Traditionally, finding biomarkers from methylation data is done by applying linear models, as these are effective at uncovering the main differences between two groups. This approach is still very powerful whenever two heterogeneous groups are being compared. More importantly, the linear models are also effective when the sample size is low. These models can also account for covariates, like age or smoking, that are two of the most influential factors on the human methylome [7]. However, diagnosis is a classification problem that can also be approached by machine learning algorithms when the sample size allows.

With the advent of modern technology, decreasing sequencing costs, and the digitalization of international cancer biobanks, we are rapidly approaching the point where AI algorithms may guide oncologists in cancer diagnosis. Such classification approaches are already emerging with well developed machine learning approaches starting to play a leading role in areas such as the medical imaging of 'melanoma' or 'prostate grade cancer' [8, 9]. The development of AI algorithms even happens in open rooms with AI competitions launched among non experts, at least in the field of cancer [10]. This is significantly driven by the quantity of data accumulated, and as such, sequencing methods have only recently started to gain much attention due to the limited number of samples available in public repositories as well as regulatory issues with sharing such data. With new initiatives that enable genomic and clinical data sharing across federated networks, such as the Beacon protocol (used as a model for the federated discovery and sharing of genomic data) [11, 12], work in this area is now in progress and will open the door to new machine learning approaches.

One of the places with a relatively high number of genomic samples available is the Genomic Data Commons data portal (GDC, https://portal.gdc.cancer.gov/). GDC represents a rich source of harmonized cancer datasets available for immediate ethical use for cancer biology and biomarkers discoveries. Most of the data are publicly available and anonymised, so therefore there is no possibility to connect individual patients with this data. GDC contains transcriptomic and genomic data as well as epigenomics information about cancers isolated from patients. Most of the epigenomics information in GDC is related to methylation profiles measured on the '450k platform' [13]. For this platform, 450 000 of the 28 million CpG sites (locations in the genetic code where a cytosine is immediately followed by a guanine) present in the human genome are measured and a value between 0 to 1 is reported for each CpG site. As the distribution of the methylation is binomial for human DNA a threshold of 0.3 can be used to classify CpGs as either methylated ($>= 0.3$) or unmethylated ($<0.3$) [14].

Artificial intelligence (AI) can operate on such data [15] although further development of this field is still required. As recently examples, a deep learning model for regression of genome-wide DNA methylation was implemented for genomic data [16]. The authors proposed a deep learning method for prediction of the genome-wide DNA methylation, in which the Methylation Regression was implemented by Convolutional Neural Networks (MRCNN). Through minimizing the continuous loss function, experiments show that their model was convergent and more precise than the previous state-of-art method (DeepCpG [17]) according to the results of the evaluation. Also, integrative analysis identifies potential DNA methylation biomarkers for 'pan-cancer' diagnosis and prognosis [18]. Unfortunately, this research used the 'ghost probe' 'cg203000343' for classification which contaminates their data. In another

recent example, machine learning methods and DNA methylation data were used to distinguish primary lung squamous cells carcinomas from head and neck metastases [19]. Other recent studies have also shown it is possible to use DNA methylation to predict disease outcome [20, 21]. All this should open methodological possibilities of using methylation data for prognosis. However, owing to the security of the genomic data it is desired to use a minimal number of genomic locations for classification purposes. Using a minimal set of genomic locations also allows for the identification of novel biomarkers.

The primary goal of our work was to develop machine learning algorithms for the diagnostic classification of DNA methylation profiles. We used data obtained from the GDC platform relating to three urological cancers; prostate, bladder and kidney. These were chosen as they each had a relatively large number of epigenetic data points available as for both cancerous tissue and normal (adjacent tissue). To develop the algorithm, we used sequential methods for feature selection and then deep learning. These methods are proposed for further independent validation in laboratories that store prostate, bladder, and kidney tissue samples and have sufficient resources.

## Results and discussion

In our AI analysis, cancer samples downloaded from GDC were more numerous than normal samples and we chose not to perform a data augmentation strategy. Using Principal Component Analysis (PCA) dimensionality reduction to reduce the data from approximately 400 000 dimensions to just two dimensions we found that the cancer samples can be visually distinguished from the normal samples with relative ease for all tissue types (Fig 1). Given that a method using PCA on 400 000 data points would be impractical in a clinical environment, this was primarily done with the intent of visualizing the data to better understand how to proceed. In particular, we wanted to see if normal samples clustered with each other, or if there were any biases in the data that needed further corrections. We also checked for samples that had been accidentally swapped during deposition to GDC. Using this visualization, we observed that the cancer data points had noticeably higher variation in methylation spread.

Using the decision tree approach we were able to select a low number CpG sites based on the criterias described in the method section while minimising the loss of any information useful for distinguishing between normal and cancer signals (Table 1, Fig 2). The CpGs were gene

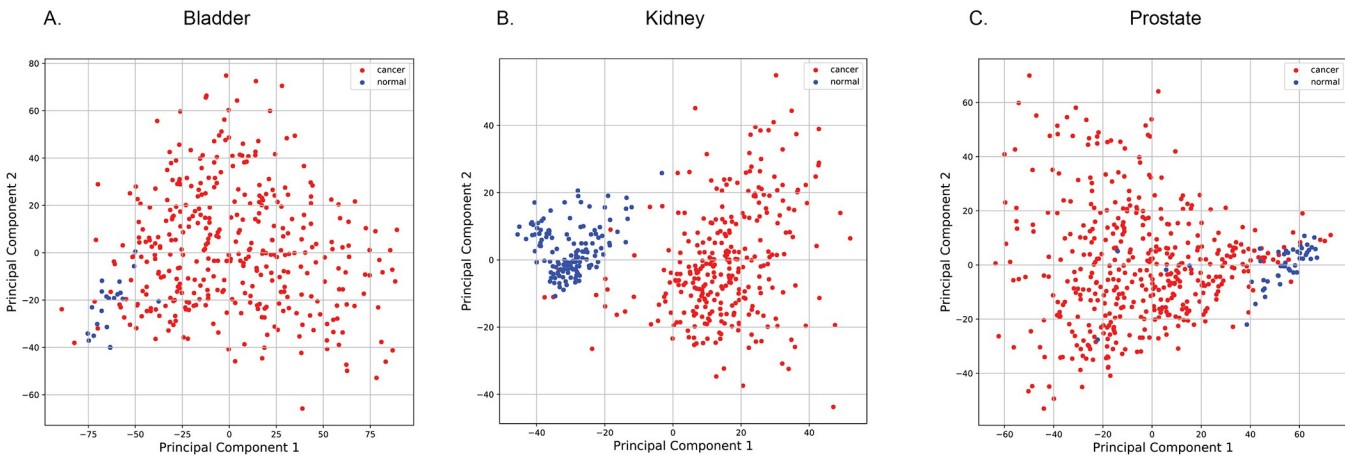

**Fig 1.** Principle component 1 and 2 for cancer (red) and normal (blue) samples for the three urinary cancer types: a) bladder, b) kidney, c) prostate. Data comes from training and validation sets.

**Table 1. CpG sites selected to distinguish between normal and cancer tissues using feature selection.** Name of CpGs as well as location is given, as well as gene annotation and position in genomic features. The order of CpGs is related to significance given in the neural network. CpGs with negative separation scores were removed.

| Type | CpG | Chr | Position | Gene annotation | Location | Separation Score |
|---|---|---|---|---|---|---|
| **Bladder** | cg04049981 | chr17 | 50817533 | - | - | 2,64 |
| | cg04349084 | chr8 | 23745164 | RP11-175E9.1 | - | 2,949 |
| | cg04449953 | chr8 | 143404769 | - | N_Shelf | 3,191 |
| | cg22746681 | chr10 | 50152871 | - | Island | 1,356 |
| | cg06667961 | chr6 | 31559916 | - | - | -16,968 |
| | cg16086237 | chr12 | 130442711 | RIMBP2 | - | 3,293 |
| | cg18997129 | chr7 | 143408757 | EPHA1;EPHA1-AS1; | - | 1,622 |
| | cg19098932 | chr1 | 2413713 | PEX10 | S_Shore | 1,235 |
| | cg19900821 | chr1 | 162389343 | - | - | 0,862 |
| | cg20696143 | chr12 | 132405765 | - | N_Shore | 2,453 |
| | cg21720373 | chr7 | 56487789 | - | S_Shelf | 3,171 |
| **Kidney** | cg00221185 | chr6 | 30684491 | AL662797.1;PPP1R18 | N_Shelf | 3,092 |
| | cg00375457 | chr11 | 5596043 | AC015691.13;HBG2;TRIM6;TRIM6-TRIM34 | - | -9,741 |
| | cg01904393 | chr12 | 51421299 | SLC4A8 | N_Shelf | 3,216 |
| | cg02578087 | chr3 | 8629675 | SSUH2 | - | 3,164 |
| | cg03272310 | chr17 | 41548202 | - | N_Shore | 6,51 |
| | cg03564506 | chr7 | 98878278 | TRRAP | N_Shore | -136,373 |
| | cg03852551 | chr6 | 167182751 | TCP10L2 | - | 3,643 |
| | cg06621027 | chr2 | 79952633 | CTNNA2 | - | 1,845 |
| | cg14204586 | chr1 | 155962067 | ARHGEF2 | - | 8,381 |
| | cg22274117 | chr6 | 16713382 | ATXN1 | - | 12,399 |
| **Prostate** | cg00183173 | chr6 | 170024144 | - | Island | -50,587 |
| | cg00333364 | chr11 | 8683255 | AC091053.1;RPL27A;SNORA45A | S_Shore | -2,998 |
| | cg00344260 | chr12 | 44876623 | NELL2 | Island | -48,392 |
| | cg00958578 | chr16 | 53715 | POLR3K;SNRNP25 | Island | -102,875 |
| | cg02049405 | chr6 | 30127488 | - | Island | 1,255 |
| | cg08206623 | chr11 | 2886104 | CDKN1C | S_Shore | 0,064 |
| | cg09691340 | chr17 | 44325514 | SLC25A39 | Island | 2,712 |
| | cg09808235 | chr1 | 2611149 | MMEL1 | - | 1,787 |
| | cg11417025 | chr7 | 16465964 | SOSTDC1 | - | 1,792 |
| | cg12265829 | chr14 | 24334816 | ADCY4; RP11-934B9.3 | Island | 1,914 |
| | cg15267232 | chr10 | 8055726 | GATA3 | Island | 2,158 |
| | cg22621867 | chr3 | 51956285 | GPR62 | Island | 1,658 |

annotated and we reported their identifiers, locations (on human genome v38), and separation score (Table 1). The CpG sites to be used for neural network classification were further refined by removing those in which the median value for the cancer and normal groups both fell on the same side of the cutoff point. Performed gene enrichment analysis, did not show any direct relationship to cancer pathways.

The final CpGs used in the neural network were subjected to clustering using the Ward clustering method and visualized (Fig 3). Bladder and kidney cancer types were generally characterized by CpGs that were hypomethylated (low methylation), while for prostate we identified hypermethylated (high methylation) sites defining cancer. We then performed ROC curve analysis on our testing set and found our neural network produced very accurate results (Fig 4).

**A. Bladder**

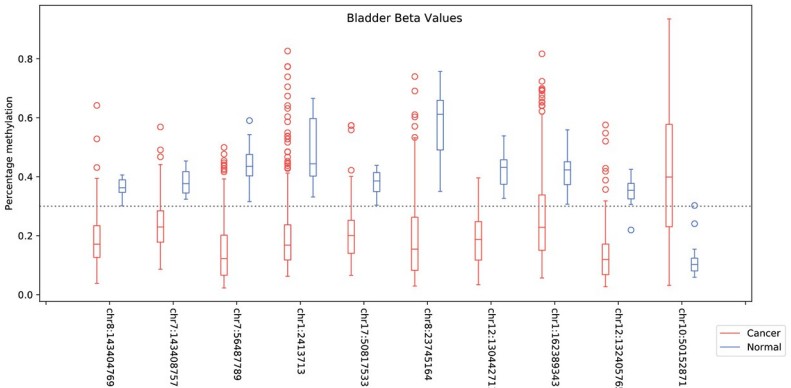

**B. Kidney**

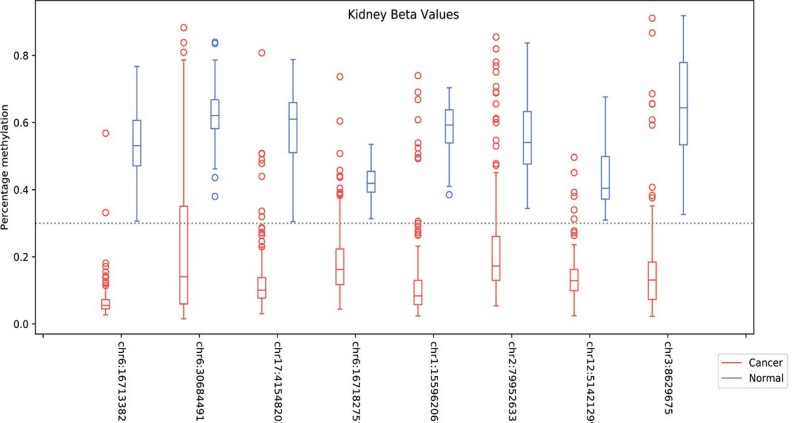

**C. Prostate**

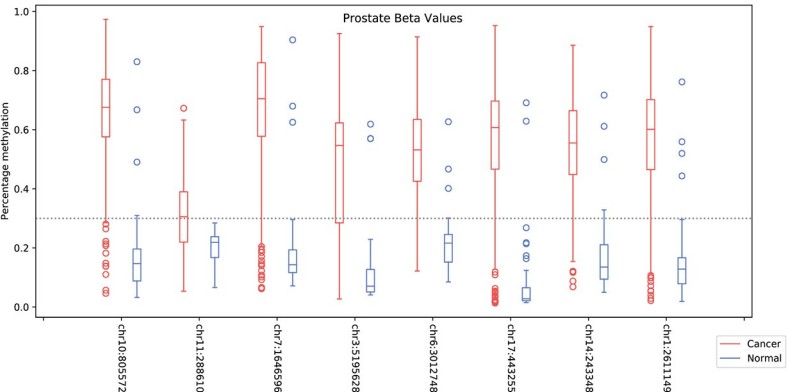

**Fig 2. Boxplot visualisation of CpG sites (on hg38 genome) methylation per group cancer (red) and normal (blue) picked by the decision tree for each tissue type and further used by the neural network.** Data for A. bladder, B. kidney and C. prostate tissue is presented. The threshold of 0.3, defining methylated or unmethylated was marked with a dashed line. Data comes from the training and validation sets.

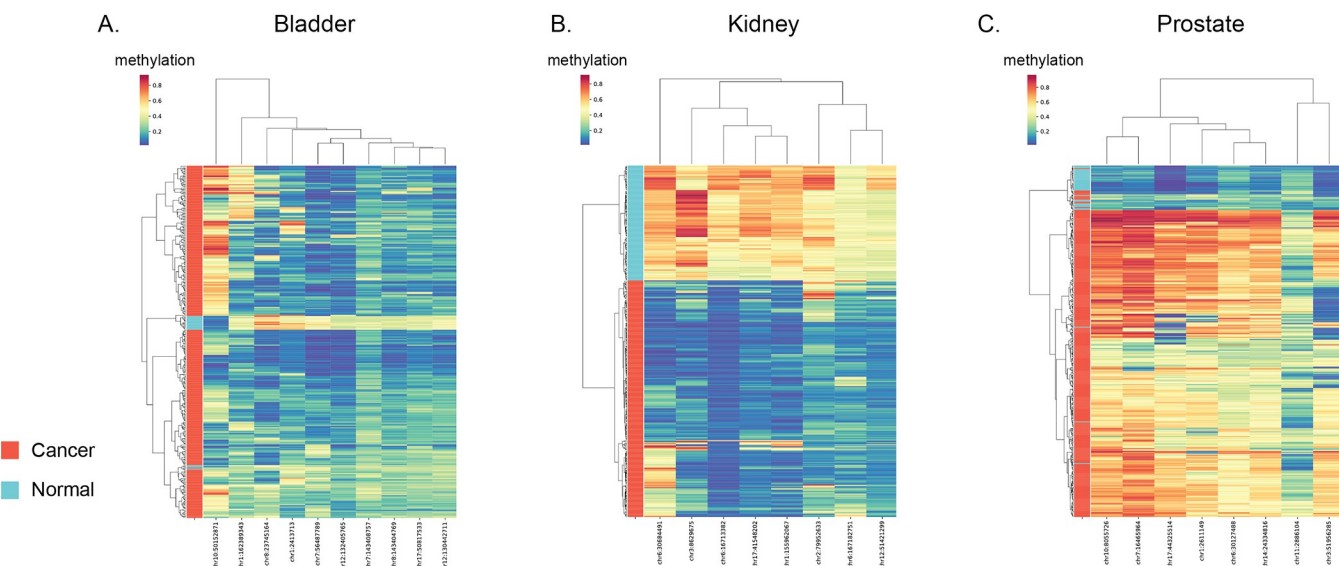

**Fig 3.** Cluster maps for A. bladder, B. kidney and C. prostate cancer. Colour scale represents methylation values. Samples are clustered by the Cancer and Normal samples and chosen CpGs. Data originates from the training and validation sets but the testing set is excluded.

## Conclusions

As novel and cost-effective genomic methods emerge and computational methods become more sophisticated, the monitoring of changes in epigenetic signatures of human DNA could be a powerful tool for finding prognostic cancer biomarkers as well as for many screening programs. Ideally, many future screening programs could be based on the epigenomic tests although this would require a very high sensitivity and specificity of these tests [15].

Traditional classification methods (such as linear models based on methylation data differences between groups) focus on areas of high variance within the methylation levels. However, they may miss interactions between points that could offer insight into the underlying biology behind developing tumours. AI methods may be more capable of identifying these interactions, and selecting CpG sites that play a role in the development of cancer in conjunction

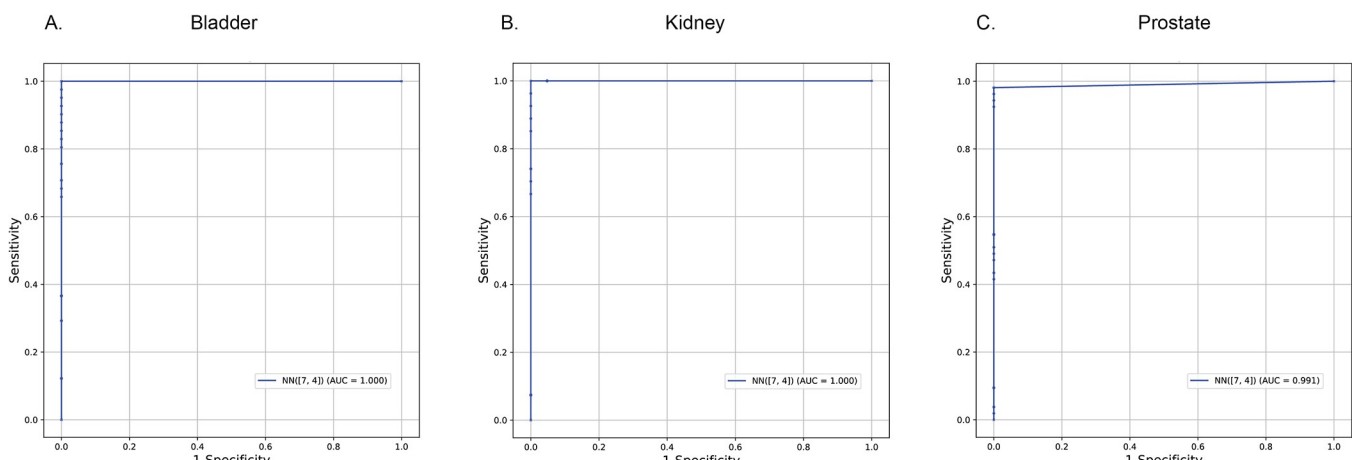

**Fig 4.** Receiver operating characteristic (ROC curves) for A. bladder, B. kidney and C. prostate cancer. First neural layer had 7 and second 4 neurons. Area under the curve (AUC) is marked. Data originates from the testing set only.

with another CpG location, but that do not necessarily display a high variance in methylation levels between cancerous and non-cancerous tissues themselves. Our approach also paves the way for the development of more AI methods that can work on methylation data, combining it with other data types, to reveal insights for which linear methods may not be adequate.

However, there is no doubt that to achieve such a goal, investment in a large number of samples with measured methylation statuses is still desired. Currently, genomic samples are scarce in public databases and their availability is hampered by data privacy issues, however the GDC platform already offers a rich source of anonymised methylation data. We used this in our work for machine learning approaches, to demonstrate the viability of our AI approach in the context of urological cancers.

Using normalized methylation values in data provided by the GDC community we have been able to identify unique signatures for three cancer types, and distinguish these from their respective normal tissues. As we were looking for major differences between samples we did not use any metadata related to survival or age of the patient. This approach is desired as it decreases the possibility of back identification of any given patients which could compromise the future use of this technique in diagnostics. Importantly, we were still able to get a clear signal for these cancers without compromising personal information.

For this study, we divided our data into methylated and unmethylated groups prior to processing, based on a threshold of 0.3. This resulted in a clear separation of the cancer and normal tissues for many CpG sites (quantified by their separation values in Table 1). Using this approach, CpGs with a high separation value are likely to be useful for designing future targeted approaches. We also performed gene enrichment analysis, which did not show any direct relationship to any known cancer pathways, suggesting that the CpG sites we found do not share a common function, including cancer.

We have also noticed that other feature engineering approaches (such as PCA) used for neural networks perform well, however many of them would require the whole methylation profile to be obtained from future patients and would dramatically increase the computational resources required. Although we believe this is possible, we have demonstrated that only few CpG sites are needed to obtain a high specificity and sensitivity when classifying patients.

In our study, a computationally demanding step was using the decision tree for feature selection but dimensionality reduction was a necessary step for successfully training the neural network. We used a simple AI architecture, where only a few neurons were used in each layer of the net, however, this was enough to train the model efficiently as shown by the validation set. As we kept the testing set completely separate until all other changes were finished we do not believe we have any overfitting and believe that the accuracy it demonstrated is representative of how our model would perform on similar external data.

There is a risk, however, that the lack of an independent, external set of the samples from other resources could have introduced bias into our results. The GDC data we used helps offset this somewhat given that it comes from multiple independent projects that have been deposited on GDC over several years. There is still a risk, however, that unaccounted for discrepancies in the way different projects handled this initial data (whether in the acquisition, handling, or processing stages) could have introduced biases into our sets. It would be beneficial to further test the conclusions we have reached on independently collected data. Unfortunately, we do not have access to any external data sources, however by freely releasing our algorithms we allow other users who have access to such data to test our approach.

The found biomarkers are related directly to the tissue where cancer was present, and compared with adjacent normal tissue. This creates the possibility that there are field effects in this study and therefore further validation is needed. This could be done in an independent cohort.

These studies were done in tissue but could be further converted to work based on blood which would be beneficial for the development of screening tests.

Our study used data from tissues samples and thus biopsies would be necessary to apply our classification algorithm, which is not an option in screening tests. As such, this approach could be improved upon but transitioning to human material that can both be unintrusively collected and has a high chance of containing information about developing cancer. Possible options include blood, urine or spit. The first approach could be done, for instance, using large prediagnostic blood biobanks, such as the Janus Serum Biobank [22]. Samples in the blood would also need deconvolution methods to further account for different cell compositions in the blood samples.

Our study only focused on three urological cancer types, however, it is possible that pre-diagnostic screening methods could be developed for other cancers using similar methods. Further studies to explore this possibility would also be beneficial.

Finally, our results provide a demonstration of the fundamental power of AI approaches and of the usefulness of methylation signatures for diagnoses, and will hopefully pave the way towards the development net of a better understanding of how epigenetics plays a role in cancers and in the design of better and more cost-effective screening methods for urological cancers without compromising patient data safety.

## Material and methods

Our code was written in Python3, using pickle, matplotlib, pylab, scipy, numpy, seaborn, and pandas as external libraries. The program was run using University of Birmingham's Blue-BEAR HPC service, which provides a High Performance Computing service to the University's research community (http://www.birmingham.ac.uk/bear). BlueBEAR HPC uses a Slurm files system for job submission.

Data was downloaded from Genomic Data Commons (GDC) Data Portal (see data availability for manifest files and our GitHub repository, github.com/bazyliszek/methAI). All methylation data were already normalized across samples by the GDC consortium and therefore were ready for further processing. Shortly, the GDC portal performs automatic data normalization with each major release of the data. Their Methylation Array Harmonization Workflow (MAHW) uses raw methylation array data for the HumanMethylation 450k platform, measuring CpG methylation as beta values, calculated from array intensities (Level 2 data) as Beta = M/(M+U). This differs from the Methylation Liftover Pipeline in that the raw methylation array data is used instead of submitted methylation beta values, and the data is processed through the SeSAMe software [23, 24]. Additionally, the analysis results from the MAHW are of higher quality than results from the Methylation Liftover Pipeline. In our work we used only HumanMethylation 450k data. SeSAMe software can remove non-detection artifacts that would otherwise survive existing pipelines, such as Y-chromosome probes. It also corrects detection failures caused by germline and somatic deletions in other DNA methylation array software by calculating the significance of signals in the methylation arrays [23]. Correcting for these artifacts allows SeSAMe to improve upon the detection calling and quality control of methylation data.

Nevertheless, to further confirm that the normalization between datasets was done correctly by GDC we plotted a histogram of the two datasets for each tissue type, and calculated the first four moments for the distributions between cancer and normal samples for each tissue type (S1 Fig) to check that there were no noticeable differences between the two groups.

The downloaded data contained a large number of missing (NA) values. This created a problem as the neural net requires numerical inputs and the way these values were handled

could adversely affect the decision tree's performance as well. On inspection, we discovered that the majority of these could be attributed to non-existent CpG sites (location was always specified as *) in the data files. However, even after removing these, there were still some NA values. Throwing out all rows or columns with NA values would have reduced our dataset too much to be a valid option. Instead, we first discarded all CpG sites in which NA values made up more than 15% of the data. We then discarded all patients in which NA values made up more than 15% of the data. CpG sites were removed first as we had more of them available so removing a CpG site caused less loss of information in our dataset than removing a patient. Still, there were a few patients with a sufficiently high ratio of NA values that it was still beneficial to remove them from the study.

After pruning out the CpG sites and patients with the most NA values, we were left with 439 patients in the bladder set, 484 in the kidney set, and 542 in the prostate set. Among these, a small number of NA values still remained in the data. We explored increasing the cutoff of NA values needed to exclude a patient or CpG site from the study to further reduce these, but rejected that out of concern for diminishing our dataset too much. We also considered replacing all NA values with a designated number (such as -1) so that the neural net would be able to distinguish them, but rejected that as well as it may have biased the neural net towards treating NA values similarly to low values, and *vis versa*. In the end, we settled on replacing all remaining NA values with the arithmetic mean value for that CpG site in the relevant tissue as this was considered the option the meant they were least likely to have a significant impact on our results.

The methylation values were then converted to binary by using a cutoff point of 0.3 as previously suggested [14]. CpG sites with methylation values higher than this were treated as methylated, while those with lower values were classified as unmethylated. This also has a biological meaning related to gene regulation.

After the preprocessing stage, the data was randomly divided into three sets. 70% of the data was assigned to the training set (307 for bladder, 339 for kidney, 379 for prostate), 20% was assigned to the validation set (88 for bladder, 97 for kidney, 108 for prostate), and the remaining 10% to the testing set (44 for bladder, 48 for kidney, 55 for prostate). The validation set was withheld and used later to optimize and compare different approaches, while the testing set was put aside and not used for any purpose other than calculating the final values in the results section. Values in the testing set were also excluded from any data visualization.

The ~400 000 dimensions (CpG sites) in our dataset was far more than a neural network could reasonably be expected to train on in a feasible amount of time. To resolve this we applied feature selection techniques to reduce the dataset to a more manageable number of dimensions. Initially we tried using Principal Component Analysis (PCA) to reduce the dimensions from ~400 000 to 10. This worked well and the initial tests with the neural network showed that it could classify the patients extremely accurately using the PCA data set. However, as the PCA set was constructed with input from 400 000 CpG sites, any test based on this procedure would require all 400 000 CpG sites to be measured for any patient. The cost of doing this means that this approach would be impractical in most clinical settings. Instead we attempted to identify those CpG sites most relevant for classifying the data. For this, we used a decision tree to identify approximately 10 of the most relevant CpG sites for dividing the data for each tissue type. These points were then used to train the neural network. The decision to focus on around 10 CpG sites was heavily based on the trade off between the cost associated with using a higher number if the test was to be used in a clinical setting, weighed against the loss of accuracy that reducing the number of CpG sites would cause. Analyzing 10 CpG sites would be relatively inexpensive compared with a 400k array, while still providing enough information for the neural network to classify the cases.

The optimal CpG locations to give the neural network would contain a list of locations that show a high variance in methylation status between the cancerous and normal groups (primary separating locations), and locations that help it classify the data in cases which are not separated by those high variance points (secondary separating locations). In order to try to get both of these, the decision tree was run to a max depth of two, with the first dividing CpG site being chosen as the one that best separated the two groups and the second and third sites (if applicable) being the ones that were best able to correct for any mistakes the first point made. After each run the sites found were added to the list and all subsequent runs of the decision tree were prevented from dividing on those locations in future. This forced the subsequent runs to find a variety of primary and secondary separating locations, increasing the number of features that the neural network could use for classification. At the end of the process, this resulted in a list of at least 10 CpG sites for each tissue type (actual number was slightly variable as the decision tree did not always return the same amount of sites, however it would keep running until it had at least 10, this resulted in 10 to 12 values being chosen at the end. We considered cutting it off at 10, however the purpose of using a decision tree to select the values instead of linear models was that the effectiveness of the CpG sites it could select may be highly dependent on each other being included.). When training the decision tree we used entropy as a measure of the confidence the system had in classifying a data point at a given position within the tree, to determine which features it would select. The Entropy at a given node was calculated accordingly: [25],

$$Entropy = -Normal*log(Normal) - Cancer*log(Cancer) \tag{1}$$

Where 'Normal' represents the fraction of the data points in that node that were taken from adjacent apparently non-cancerous tissue, and 'Cancer' is the fraction of the data points in that node that were taken from visibly cancerous tissue. This equation creates a curve that is at its minimum (0) when the data is fully divided and maximum (1) when the data is evenly mixed.

When training the decision tree, we found that a depth of 2 was sufficient to classify the data in >90% of cases. Running at a max depth of 2 gave us a mix of locations with high variance between the cancer and normal tissue types, and locations that were good at fine tuning the details and correcting those data points that the first locations failed to classify correctly.

The CpG lists from the decision tree were further narrowed down by excluding any in which the median methylation for each dataset fell on the same side of the cutoff point. This was done to reduce chances of the neural network basing a decision on unreliable or outlying data. The separation score (a rough estimate of how distant the two medians of the two groups were from each other's side of the cutoff point) was calculated using the formula:

$$SeparationScore = -\frac{CancerMedian - cutoff}{CancerSD} * \frac{NormalMedian - cutoff}{NormalSD} \tag{2}$$

Negative values occur when both groups have a median on the same side of the cutoff point and were excluded from the rest of the study. The values were recorded (Table 1). The final list of CpG locations was used to train three neural networks for each tissue type. A sigmoid activation function (S(x)) was chosen to calculate the output of each neuron, due to the ease of calculating the gradient for back propagation and the fact that it gives bounded outputs for all potential inputs.

$$S(x) = \frac{1}{1 + e^{-x}} \tag{3}$$

Where e is Euler's number and x is the sum of all weighted inputs for a given neuron. The

first neural network had 7 neurons in the first hidden layer and 4 in the second hidden layer. The second neural network had 10 neurons in both hidden layers. The third neural network had three hidden layers containing 5, 4, and 3 neurons respectively. All three architectures also used a learning rate function that decreased slowly over time. These architectures and the initial learning rate were chosen based on our experience training other neural nets on similar data sets, with the intent to use the validation set to optimize these parameters. However, this turned out to be unnecessary as all three structures reached the point where they were able to perfectly, or near perfectly, classify all patients in the validation set on all tissue types. This resulted in an Area Under the Curve of, or very close to, 1 for all structures over all tissue types during Receiver operating characteristic (ROC) analysis. As such, no further adjustments were needed. As our ROC curves produced such high values for each tissue type a larger testing set is likely needed to get a more accurate assessment on the neural nets actual performance. All we can say from the current results is that it is likely very high for both bladder and kidney tissue. To check if the CpGs used had any known biological functions (for instance known cancer pathway) we have used webtool Enrichr was used to find out if there is any association between genes that were in approximation of the found CpGs and their functions (https://maayanlab.cloud/Enrichr/) [26].

## Supporting information

**S1 Fig.**
(DOCX)

## Acknowledgments

The computations described in this paper were performed using the University of Birmingham's BlueBEAR HPC service, which provides a High Performance Computing service to the University's research community. Additional computations were performed on resources provided by Sigma2—the National Infrastructure for High Performance Computing and Data Storage in Norway (project number NN8039K). We thank the GDC help desk for the data and for assistance given related to retrieving the data from their servers.

## Author Contributions

**Data curation:** Marcin W. Wojewodzic, Jan P. Lavender.

**Formal analysis:** Marcin W. Wojewodzic, Jan P. Lavender.

**Funding acquisition:** Marcin W. Wojewodzic.

**Investigation:** Marcin W. Wojewodzic, Jan P. Lavender.

**Methodology:** Marcin W. Wojewodzic.

**Project administration:** Marcin W. Wojewodzic.

**Resources:** Marcin W. Wojewodzic.

**Software:** Marcin W. Wojewodzic, Jan P. Lavender.

**Visualization:** Jan P. Lavender.

**Writing – original draft:** Marcin W. Wojewodzic, Jan P. Lavender.

**Writing – review & editing:** Marcin W. Wojewodzic.

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
