## [Decision Letter · Decision Letter 0]

13 May 2024

PONE-D-23-40540Diagnostic classification based on DNA methylation profiles using sequential machine learning approaches.PLOS ONE

Dear Dr. Marcin,

Thank you for submitting your manuscript to PLOS ONE. After careful consideration, we feel that it has merit but does not fully meet PLOS ONE’s publication criteria as it currently stands. Therefore, we invite you to submit a revised version of the manuscript that addresses the points raised during the review process. **The manuscript has great potential against the background of current attempts to find discriminatory markers for distinguishment of malignant entities. I find the approach regarding the definition of biomarkers or particularly biomarker signatures based on the application of machine learning methods to DNA methylation data intriguing. However, the reviewers have expressed some concerns regarding particularly methodological aspects. Some issues have to be explained in more detail. Although I decided for “major revision” I am very confident that a satisfactory based on the reviewers´ comments is possible.**

Please submit your revised manuscript by Jun 27 2024 11:59PM. If you will need more time than this to complete your revisions, please reply to this message or contact the journal office at plosone@plos.org. Please include the following items when submitting your revised manuscript:A rebuttal letter that responds to each point raised by the academic editor and reviewer(s). You should upload this letter as a separate file labeled 'Response to Reviewers'.A marked-up copy of your manuscript that highlights changes made to the original version. You should upload this as a separate file labeled 'Revised Manuscript with Track Changes'.An unmarked version of your revised paper without tracked changes. You should upload this as a separate file labeled 'Manuscript'.

We look forward to receiving your revised manuscript.

Kind regards,

Marc Reismann, MD, PhD

Academic Editor

PLOS ONE

Journal Requirements:

7. We notice that your supplementary figure is included in the manuscript file. Please remove and upload it with the file type 'Supporting Information'. Please ensure that each Supporting Information file has a legend listed in the manuscript after the references list.

Reviewers' comments:

Reviewer's Responses to Questions

**Comments to the Author**

1. Is the manuscript technically sound, and do the data support the conclusions?

Reviewer #1: Partly

Reviewer #2: Partly

2. Has the statistical analysis been performed appropriately and rigorously? 

Reviewer #1: N/A

Reviewer #2: Yes

3. Have the authors made all data underlying the findings in their manuscript fully available?

Reviewer #1: Yes

Reviewer #2: Yes

4. Is the manuscript presented in an intelligible fashion and written in standard English?

Reviewer #1: Yes

Reviewer #2: Yes

5. Review Comments to the Author

**Reviewer #1**: The core concept this manuscript attempts to address is discrimination between cancer and adjacent normal tissue by methylation profiling, in this instance via Illumina 450K array. 3 urological cancer datasets are used as an example. The authors use a machine learning model to extract CpG sites that differentiate between cancer and normal in each tumour type, and use these to construct a classification model.

This manuscript appears mostly technically sound in terms of the implementation of the authors chosen methods, although as with any bioinformatics analysis or machine learning model it is impossible to definitively assess methodological rigour without viewing the underlying code (which is not public/viewable at the time of review). Nevertheless, the descriptions given in the methods section are sufficient for an adequate representation of the rationale underlying this particular implementation, if not replication. Machine learning-based classification of tumours is a well established approach (e.g., the MNP CNS tumour classification system) and though the model in this manuscript uses a different algorithm, the principle of classification via methylation is sound. The manuscript is written in an intelligible manner with a few grammatical mistakes but nothing that should impede a proper understanding of the written language.

I have some technical/rationale queries for the authors after reading the manuscript:

1. I understand data was previously normalised across samples by GDC before retrieval by the authors. Did this normalisation include removal of CpGs associated with the X and Y chromosomes, probes with known mismatches, and probes adjacent to SNPs? If not, did the authors remove these CpGs as part of their preprocessing? This is not clear from the methods.

2. Regarding principal component analyses and dimensionality reduction of the data: The separation of cancer and normal samples here is a little lackluster compared to what I usually see in cancer vs normal analyses. Feeding the entire methylation array into a PCA is a rather curious approach. Many of the probes on the array will demonstrate no or very little variance between cancer and normal tissue, but will introduce significant noise to the final resolving power of the analysis. I find that selecting a subset of ~10,000 probes with high variability across the cohort (e.g., ranking probes by median absolute deviation) dramatically improves separation of biological groups when performing PCA/TSNE while capturing almost all biological variability. Similar subsetting steps prior to PCA are a fairly common step in most discrimination/class discovery analyses using methylation data. I mention this primarily as it may improve segregation on the PCA analyses the authors performed.

3. If the goal is to identify a minimal selection of CpGs that can act as markers to reliably discriminate between cancer and normal, why not perform a simple differential methylation analysis of cancer v normal for each tumour type? This would give a list of differentially methylated CpGs. Sites with a consistently large difference between cancer and normal tissue could then be chosen as standalone markers for validation or combined and incorporated into a classification model. It seems this approach would have more biological grounding, and would likely reflect major underlying biological changes between tumour/normal. While the machine learning approach to feature selection is sophisticated, I wonder about the added value of using a rather convoluted method to achieve a relatively simple goal (binary classification).

**Reviewer #2:** This is an interesting manuscript that investigates DNA methylation patterns in three type of cancer using archived data from the Genomic Data Commons. A few minor issues in the approach should be addressed prior to publication.

Under Methods:

Add specific detail in the first paragraph. The manuscript currently states, “We used ‘Python’ as the primary programming language for this project with libraries such as ‘pickle’ and ‘matplotlib’.” Please include all programming languages and libraries used.

Explain how data are determined to be “NA” in Illumina 450k data. Beyond the absence of a specific CpG, why were specific measurements designated NA?

In the third paragraph, it is state, “We explored a few options for how to deal with these, and settled on replacing all remaining NA values with the average value for that CpG site in the relevant tissue as this was considered the option least likely to have an impact on our results.” Specify the “average” used (was this the arithmetic mean? Geometric mean? Median?). Describe all the options that were explored. In the Results, provide the data that were used to make an objective determination that the average would be used.

In paragraph 6, the statement “… the initial tests with the neural network showed that it could classify the patients extremely accurately using the PCA data set (data not shown)” is not very useful. Please provide the data and specify the accuracy of the classification.

Also in paragraph 6, it is stated that “… any test based on this procedure would require all 400 000 CpG sites to be measured for any patient. This would be impractical and unreliable in clinical settings …” Please provide an explanation as to why the approach would be unreliable.

Also in paragraph 6, it is stated that “… we used a decision tree to identify approximately 10 of the most relevant CpG sites …”. Please explain why the target value of 10 was chosen. Was this a subjective choice or based on prior data or experience?

In paragraph 7, it is stated that the “actual number was slightly variable ...” please provide the actual range of values.

In paragraph 8, the explanation of Entropy as “a measure of how mixed the data is calculated accordingly” is confusing. Please provide a more specific explanation.

In paragraph 9, the statement, “we found that a depth of 2 was almost always sufficient to classify the data” should be replaced with actual classification data. What does “almost always sufficient” objectively mean?

In paragraph 12, consider replacing the term “chosen based on intuition.” Were initial learning rates selected based on prior experience with similar systems?

Finally, a specific statement about the limitations of the available data sets and analyses that could affect performance of the approach is needed. For example, variability in biological sample acquisition, storage, or processing could within and between studies could impact the analysis. Also, contributions from biological (e.g. grade of cancer) versus technical (e.g., low probe reliability) sources of variance could affect interpretation.

6. PLOS authors have the option to publish the peer review history of their article (what does this mean?). If published, this will include your full peer review and any attached files.

Reviewer #1: No

Reviewer #2: No

---

## [Author Response · Author response to Decision Letter 0]

11 Jun 2024

Reviewers' comments:

Reviewer's Responses to Questions

Comments to the Author

1. Is the manuscript technically sound, and do the data support the conclusions?

Reviewer #1: Partly

Reviewer #2: Partly

2. Has the statistical analysis been performed appropriately and rigorously?

Reviewer #1: N/A

Reviewer #2: Yes

3. Have the authors made all data underlying the findings in their manuscript fully available?

Reviewer #1: Yes

Reviewer #2: Yes

4. Is the manuscript presented in an intelligible fashion and written in standard English?

Reviewer #1: Yes

Reviewer #2: Yes

5. Review Comments to the Author

Reviewer #1: The core concept this manuscript attempts to address is discrimination between cancer and adjacent normal tissue by methylation profiling, in this instance via Illumina 450K array. 3 urological cancer datasets are used as an example. The authors use a machine learning model to extract CpG sites that differentiate between cancer and normal in each tumour type, and use these to construct a classification model.

This manuscript appears mostly technically sound in terms of the implementation of the authors chosen methods, although as with any bioinformatics analysis or machine learning model it is impossible to definitively assess methodological rigour without viewing the underlying code (which is not public/viewable at the time of review). Nevertheless, the descriptions given in the methods section are sufficient for an adequate representation of the rationale underlying this particular implementation, if not replication. Machine learning-based classification of tumours is a well established approach (e.g., the MNP CNS tumour classification system) and though the model in this manuscript uses a different algorithm, the principle of classification via methylation is sound. The manuscript is written in an intelligible manner with a few grammatical mistakes but nothing that should impede a proper understanding of the written language.

I have some technical/rationale queries for the authors after reading the manuscript:

** MW: Thank you for your constructive comments. We have now opened the code in GitHub according to FAIR principle and the code can be now accessed under the following link: https://github.com/bazyliszek/methAI as well in Zenodo. We made a change in the main text “The source code is available in GitHub repository (https://github.com/bazyliszek/methAI), and in Zenodo under DOI: 10.5281/zenodo.11530987” and can be cited Wojewodzic, M. W., & Lavender, J. P. (2024). Diagnostic classification based on DNA methylation profiles using sequential machine learning approaches (Version v0.1.0-alpha-2) [Computer software]. https://doi.org/10.5281/zenodo.11530987

1. I understand data was previously normalised across samples by GDC before retrieval by the authors. Did this normalisation include removal of CpGs associated with the X and Y chromosomes, probes with known mismatches, and probes adjacent to SNPs? If not, did the authors remove these CpGs as part of their preprocessing? This is not clear from the methods.

*** MW: Thank you for your comments. We have now improved this in our description of the method, as we understand this could be beneficial for the reader to better understand our data sources and how the GDC processed the data prior to our usage. 

We have consulted the author of the GDC pipeline to better understand the process by which probes are normalised and how probes with known mismatches or those adjacent to SNPs/repeats are discarded, as described in this paper https://academic.oup.com/nar/article/45/4/e22/2290930. 

We have described some of these steps in our manuscript.

Shortly, GDC portal automatically normalised data across samples at every major release of the data. The Methylation Array Harmonization Workflow, used by GDC, uses raw methylation array data from different generations of Illumina Infinium DNA methylation arrays: Human Methylation 27 (HM27), HumanMethylation 450 (HM450) and EPIC platforms, to measure the level of methylation at known CpG sites as beta values, calculated from array intensities (Level 2 data) as Beta = M/(M+U). This differs from the Methylation Liftover Pipeline in that the raw methylation array data is used instead of submitted methylation beta values, and the data is processed through the software package SeSAMe [1]. In our work we used only HumanMethylation 450 (HM450) data.

SeSAMe effectively removes non-detection artefacts that survive existing pipelines, including Y-chromosome probes. SeSAMe corrects to detection failures that occur in other DNA methylation array software commonly due to germline and somatic deletions by utilizing a novel way to calculate the significance of detected signals in methylation arrays. By correcting for these artifacts as well as other improvements to DNA methylation data processing, SeSAMe improves upon detection calling and quality control of processed DNA methylation data. SeSAMe output files include: two Masked Methylation Array IDAT files, one for each color channel, that contains channel data from a raw methylation array after masking potential genotyping information; and a subsequent Methylation Beta Value TXT file derived from the two Masked Methylation Array IDAT files, that displays the calculated methylation beta value for CpG sites.

We also included following publication in the paper.

[1]. Zhou, Wanding, Triche Timothy J., Laird Peter W. and Shen Hui. "SeSAMe: Reducing artifactual detection of DNA methylation by Infinium BeadChips in genomic deletions." Nucleic Acids Research. (2018): doi: 10.1093/nar/gky691

[2]. Zhou, Wanding, Laird Peter L., and Hui Shen. "Comprehensive characterization, annotation and innovative use of Infinium DNA methylation BeadChip probes." Nucleic Acids Research. (2016): doi: 10.1093/nar/gkw967

2. Regarding principal component analyses and dimensionality reduction of the data: The separation of cancer and normal samples here is a little lackluster compared to what I usually see in cancer vs normal analyses. Feeding the entire methylation array into a PCA is a rather curious approach. Many of the probes on the array will demonstrate no or very little variance between cancer and normal tissue, but will introduce significant noise to the final resolving power of the analysis. I find that selecting a subset of ~10,000 probes with high variability across the cohort (e.g., ranking probes by median absolute deviation) dramatically improves separation of biological groups when performing PCA/TSNE while capturing almost all biological variability. Similar subsetting steps prior to PCA are a fairly common step in most discrimination/class discovery analyses using methylation data. I mention this primarily as it may improve segregation on the PCA analyses the authors performed.

*** MW: Thank you for this comment. We agree with the reviewer, however at this stage of analysis we just wanted to visualise how the data clustered before performing feature selection later. In particular, we wanted to see if normal samples clustered with each other, or if there were any biases in the data that needed further corrections. This would also identify samples that were accidentally swapped during deposition to GDC. As we can notice there is a separation between groups on PC1 axes.

Initially we tried using Principal Component Analysis (PCA) to reduce the dimensions from ~400 000 to 10. However, because these 10 features were generated from 400 000, using them to classify a patient would require analysing those 400 000 CpG sites for the patient. This would be impractical in most real-world settings. As such, we instead attempted to find a small number of CpG sites that could be used without PCA. 

We have now implemented this into our manuscript in the results section.

3. If the goal is to identify a minimal selection of CpGs that can act as markers to reliably discriminate between cancer and normal, why not perform a simple differential methylation analysis of cancer v normal for each tumour type? This would give a list of differentially methylated CpGs. Sites with a consistently large difference between cancer and normal tissue could then be chosen as standalone markers for validation or combined and incorporated into a classification model. It seems this approach would have more biological grounding, and would likely reflect major underlying biological changes between tumour/normal. While the machine learning approach to feature selection is sophisticated, I wonder about the added value of using a rather convoluted method to achieve a relatively simple goal (binary classification).

*** MW: Thank you for this comment. Yes, we are aware of that and in fact we have discussed this in the introduction: “Traditionally, finding biomarkers from methylation data is done by applying linear models, as these are effective at uncovering the main differences between two groups. However, diagnosis is a classification problem that can also be approached by machine learning algorithms when the sample size allows”. 

Yet, to account for this comment we included also following text in the discussion: “Traditional classification methods (such as linear models based on methylation data differences between groups) focus on areas of high variance within the methylation levels. However, they may miss interactions between points that could offer insight into the underlying biology behind developing tumours. AI methods may be more capable of identifying these interactions, and selecting CpG sites that play a role in the development of cancer in conjunction with another CpG location, but that do not necessarily display a high variance in methylation levels between cancerous and non-cancerous tissues themselves.”

Our approach also paves the way for the development of more AI methods that can work on methylation data and reveal insights for which linear methods may not be adequate. 

Reviewer #2: This is an interesting manuscript that investigates DNA methylation patterns in three type of cancer using archived data from the Genomic Data Commons. A few minor issues in the approach should be addressed prior to publication.

*** MW: Thank you very much for the interest in our paper and the feedback we got.

Under Methods:

Add specific detail in the first paragraph. The manuscript currently states, “We used ‘Python’ as the primary programming language for this project with libraries such as ‘pickle’ and ‘matplotlib’.” Please include all programming languages and libraries used.

*** MW: Thank you. We have not implemented versions of the packages. We added missing packages we used in the text: pylab, scipy, numpy, seaborn, pandas. 

Explain how data are determined to be “NA” in Illumina 450k data. Beyond the absence of a specific CpG, why were specific measurements designated NA?

*** MW: The NA values originate from the Illumina platform and are associated with low quality probes, or chromosome Y. We have now mentioned this in the method section and how preprocessing is done by GDC. 

In the third paragraph, it is state, “We explored a few options for how to deal with these, and settled on replacing all remaining NA values with the average value for that CpG site in the relevant tissue as this was considered the option least likely to have an impact on our results.” Specify the “average” used (was this the arithmetic mean? Geometric mean? Median?). Describe all the options that were explored. In the Results, provide the data that were used to make an objective determination that the average would be used.

*** MW: We have used the arithmetic mean of all non-NA data points at that CpG site. We have also described in detail which alternative methods we explored and rejected (the same paragraph). 

In paragraph 6, the statement “… the initial tests with the neural network showed that it could classify the patients extremely accurately using the PCA data set (data not shown)” is not very useful. Please provide the data and specify the accuracy of the classification.

Also in paragraph 6, it is stated that “… any test based on this procedure would require all 400 000 CpG sites to be measured for any patient. This would be impractical and unreliable in clinical settings …” Please provide an explanation as to why the approach would be unreliable.

*** MW: Unfortunately, we no longer have the data showing the accuracy of the 400 000 classification approach given that this method was rejected as unusable for other reasons. We have expanded a little upon the rationale behind this rejection.

The 400 000 approach was rejected as unreliable on the basis that such a large quantity of data being collected would increase the rate of errors being introduced to the set. However, in responding to recommendations to the other reviewer we removed the statement that this would be unreliable and focused on the impracticality and expense of attempting to use a 400 000 CpG method in a clinical setting. As such, it no longer refers to the approach as unreliable.

Also in paragraph 6, it is stated that “… we used a decision tree to identify approximately 10 of the most relevant CpG sites …”. Please explain why the target value of 10 was chosen. Was this a subjective choice or based on prior data or experience?

*** MW: The decision to focus on around 10 CpG sites was heavily based on the trade-off between the cost associated with using a higher number if the test was to be used in a clinical setting, weighed against the loss of accuracy that reducing the number of CpG sites would cause. Analysing 10 CpG sites is relatively inexpensive but involves very little information loss. We have implemented this in the paper now. 

In paragraph 7, it is stated that the “actual number was slightly variable ...” please provide the actual range of values.

*** MW: Thank you. We have added the actual number of return values (10 to 12). We have now implemented this to the article (paragraph 7) and added justification.

In paragraph 8, the explanation of Entropy as “a measure of how mixed the data is calculated accordingly” is confusing. Please provide a more specific explanation.

MW: We have improved this further. "Entropy is a measure of the confidence with which the system can classify a data point at a given position in the tree" to the paper. (Paragraph 8)

In paragraph 9, the statement, “we found that a depth of 2 was almost always sufficient to classify the data” should be replaced with actual classification data. What does “almost always sufficient” objectively mean?

*** MW: Depth of 2 was sufficient for classification for more 

---

## [Decision Letter · Decision Letter 1]

11 Jul 2024

Diagnostic classification based on DNA methylation profiles using sequential machine learning approaches.

PONE-D-23-40540R1

Dear Dr. Marcin,

We’re pleased to inform you that your manuscript has been judged scientifically suitable for publication and will be formally accepted for publication once it meets all outstanding technical requirements.

Kind regards,

Marc Reismann, MD, PhD

Academic Editor

PLOS ONE

Reviewers' comments:

Reviewer's Responses to Questions

**Comments to the Author**

1. If the authors have adequately addressed your comments raised in a previous round of review and you feel that this manuscript is now acceptable for publication, you may indicate that here to bypass the “Comments to the Author” section, enter your conflict of interest statement in the “Confidential to Editor” section, and submit your "Accept" recommendation.

Reviewer #1: All comments have been addressed

Reviewer #2: All comments have been addressed

2. Is the manuscript technically sound, and do the data support the conclusions?

Reviewer #1: Yes

Reviewer #2: Yes

3. Has the statistical analysis been performed appropriately and rigorously? 

Reviewer #1: Yes

Reviewer #2: Yes

4. Have the authors made all data underlying the findings in their manuscript fully available?

Reviewer #1: Yes

Reviewer #2: Yes

5. Is the manuscript presented in an intelligible fashion and written in standard English?

Reviewer #1: Yes

Reviewer #2: Yes

6. Review Comments to the Author

Reviewer #1: The authors have made suitable efforts to engage with the previous review comments and have made modifications to the manuscript to add clarity and detail where suggested. The authors have defended their analysis and, while I find it a slightly curious approach, there is no indication that this is unsound or inappropriate. My main minor suggestion to improve the manuscript would be to add further descriptive prose to the results section. At present, after the first paragraph, it is a little lackluster and doesn't give much indication of the results (e.g., "Performed gene enrichment analysis, did not show any direct relationship to cancer pathways."; "...performed ROC curve analysis on our testing set and found our neural network produced very accurate results."). While the associated figures and tables add some context, description in the text would be useful to highlight specific details e.g, accuracy metrics etc.

Reviewer #2: The authors have addressed my concerns. The manuscript has been updated and is now suitable for publication.

7. PLOS authors have the option to publish the peer review history of their article (what does this mean?). If published, this will include your full peer review and any attached files.

Reviewer #1: No

Reviewer #2: No

---

## [Editor Report · Acceptance letter]

16 Jul 2024

PONE-D-23-40540R1 

PLOS ONE

Dear Dr. Marcin, 

I'm pleased to inform you that your manuscript has been deemed suitable for publication in PLOS ONE. Congratulations! Your manuscript is now being handed over to our production team.

Kind regards, 

on behalf of

Dr. Marc Reismann 

Academic Editor

PLOS ONE